# Probing the Particle Spectrum of Nature with Evaporating Black Holes

Michael J. Baker[1, *] and Andrea Thamm[2, †]

[1] *ARC Centre of Excellence for Dark Matter Particle Physics,*
*School of Physics, The University of Melbourne, Victoria 3010, Australia*
[2] *School of Physics, The University of Melbourne, Victoria 3010, Australia*
(Dated: August 3, 2021)

Photons radiated from an evaporating black hole in principle provide complete information on the particle spectrum of nature up to the Planck scale. If an evaporating black hole were to be observed, it would open a unique window onto models beyond the Standard Model of particle physics. To demonstrate this, we compute the limits that could be placed on the size of a dark sector. We find that observation of an evaporating black hole at a distance of 0.01 parsecs could probe dark sector models containing one or more copies of the Standard Model particles, with any mass scale up to 300 TeV.

**Introduction** – Determining the particle spectrum of nature is one of the fundamental goals of physics. The last 120 years have seen a huge advance in our understanding of the elementary particles, from J.J. Thomson's discovery of the electron in 1897 [1] to the discovery of the Higgs boson at CERN in 2012 [2, 3], completing the Standard Model (SM) of particle physics.

The last 100 years have also seen a huge advance in our understanding of black holes (BH), from Schwarzschild [4] and Droste's [5] exact solutions to the Einstein field equations, which would prove to describe the simplest black holes, in 1916 to the 2016 observation of gravitational waves from a binary black hole merger by the LIGO and Virgo Collaborations [6]. This was quickly followed by the first direct image of a black hole by the Event Horizon Telescope [7].

In 1974 Hawking [8, 9] combined arguments from quantum mechanics and general relativity to predict that black holes should radiate particles, so-called Hawking radiation, and lose mass. The emission is approximately black-body, with a temperature that is inversely proportional to the black hole's mass. As the black hole radiates, it loses mass and heats up, leading to a runaway evaporation process. While the solar mass and supermassive black holes already observed will not evaporate any time soon, primordial black holes with masses around $10^{15}$ g, which may have been produced in the early universe [10–33], would be evaporating today (see, e.g., refs. [34–39] for recent reviews of primordial black holes). Although there is not yet any clear evidence of evaporating black holes (EBHs), they have been invoked to explain, e.g., fast gamma ray bursts [40], antimatter in cosmic rays [41–43], and the galactic gamma ray background [44].

Evaporating black holes predominantly radiate all elementary particles with a mass less than their temperature. When the temperature rises above a particle mass threshold, a new radiation process becomes unsuppressed, the black hole loses mass at a faster rate, and the temperature increases at a faster rate. This continues until the temperature reaches the Planck scale, at which point quantum gravity effects may become important. Since photons are massless they are always emitted by evaporating black holes, with an energy similar to the black hole temperature. In addition, other radiated particles may also produce photons after their emission. In this way, the photon signal from an evaporating black hole encodes detailed information about the evaporation rate and the complete particle spectrum.

Experiments such as the HAWC Observatory are actively searching for evaporating black holes. In this work we consider what information could be obtained from an observation in practice, and the extent to which Beyond the Standard Model (BSM) scenarios could be probed. As an illustrative scenario we consider dark sector models. Dark sector models are strongly motivated by the observation of dark matter, but at present there are no known general probes of the extent of the dark sector.

While the impact of non-Standard Model physics on black hole evaporation has been discussed in the literature, this has predominantly focused on Hagedorn-type models [45], e.g. refs. [40, 46], which have now been superseded by quantum chromodynamics or BSM particle production, e.g. refs. [47–66]. To our knowledge, the impact of contemporary BSM models on the observed signal from an evaporating black hole is almost completely unexplored, with the exception of ref. [67] which contains a limited analysis in the case of a single squark.

**Formalism** – We now discuss the theoretical framework of BH evaporation, calculate the resulting photon spectra, and provide relevant details of the HAWC observatory (our example experiment) and

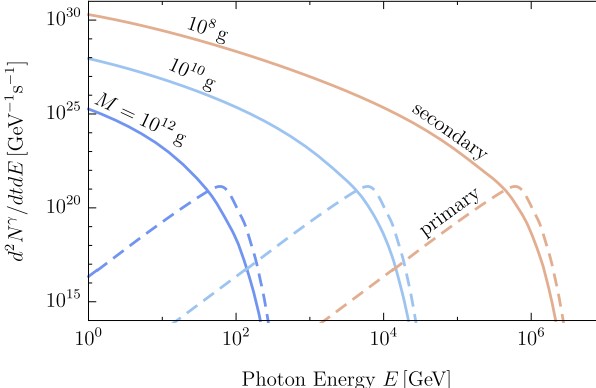

FIG. 1. The primary (dashed) and secondary (solid) photon spectra at BH masses $M = (10^{12}, 10^{10}, 10^8)\,$g. In the SM, this corresponds to $\tau = (5 \times 10^8, 4 \times 10^2, 4 \times 10^{-4})\,$s where $\tau$ is the remaining lifetime of the EBH.

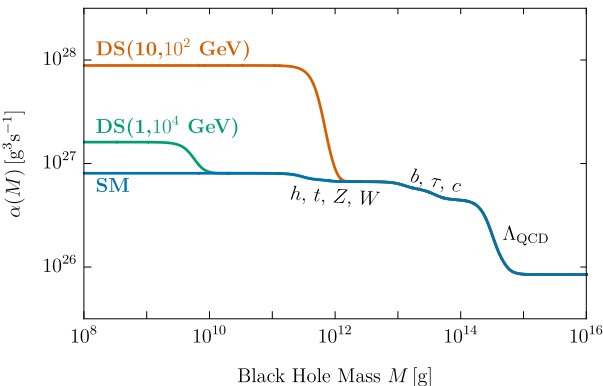

FIG. 2. The function $\alpha(M)$, which accounts for all directly emitted particle species, for the SM and two dark sector models (see text for details). The SM particle labels show the particles responsible for the thresholds. Light quarks and gluons are only radiated above $\Lambda_{\rm QCD}$.

the astrophysical gamma ray background.

BHs can be completely characterised by their mass, charge and angular momentum. However, EBHs radiate charge and angular momentum faster than they radiate mass [68–73]. As such, we can assume that EBHs, at the end of their lives, are Schwarzschild black holes, which are uncharged and non-rotating. Schwarzschild BHs are then completely characterised by their mass, $M$.

Working in units where $\hbar = c = \kappa_{\rm B} = 1$, the temperature of a BH is given by [8, 9]

$$ T = \frac{1}{8\pi\,GM}\,, \tag{1} $$

where $G$ is the gravitational constant. BHs heavier than $\sim 10^{-8} M_\odot \sim 10^{26}\,$g are colder than the CMB and are absorbing CMB photons, so are gaining mass [74, 75]. Lighter BHs, on the other hand, radiate particles of energy $E$ at the rate [8, 9]

$$ \frac{d^2 N_{\rm P}^i}{dtdE} = \frac{n_{\rm dof}^i\,\Gamma^i(M,E)}{2\pi(e^{E/T} \pm 1)}\,, \tag{2} $$

where $n_{\rm dof}^i$ is the number of degrees of freedom of particle $i$, $+$ $(-)$ corresponds to fermions (bosons) and $\Gamma^i(M,E)$ is a greybody factor that for a Schwarzschild black hole depends on the spin and energy of the radiated particle and on the mass of the black hole. The greybody factor can be calculated by solving a Schrödinger-like wave equation and finding the transmission coefficient of the solution from the BH horizon to infinity. We take the values made publicly available in ref. [76]. For $E \gg m^i$, where $m^i$ is the mass of the radiated particle, $\Gamma^i$ can be written as a function of the dimensionless quantity $x = 8\pi\,GME$. Although at $E \sim m^i$ there is a

correction to this approximation [73], particles with $E \sim m^i$ only make up a small proportion of the radiated particles and we neglect this effect. At $E < m^i$, $\Gamma^i = 0$. The greybody factor then only depends on the particle spin and $x$. The primary photon spectra for a range of BH masses are shown in fig. 1.

Conservation of energy implies that as the BH radiates, it must lose mass. The BH mass evolves according to [72]

$$ \frac{dM}{dt} = -\,\frac{\alpha(M)}{M^2}\,, \tag{3} $$

where

$$ \alpha(M) = M^2 \sum_i \int_0^\infty \frac{d^2 N_{\rm P}}{dtdE}(M,E)E\,dE\,. \tag{4} $$

and the sum is over all particle species. All fundamental degrees of freedom present in nature with a de Broglie wavelength of the order of the black hole size are radiated [77], so contribute to $\alpha(M)$. Note in particular that $\alpha(M)$ is independent of the particle's non-gravitational interaction strengths. In fig. 2 we show $\alpha(M)$ for the SM in blue.

Although EBHs emit all particles, only stable particles can reach the earth to be observed, and only uncharged particles will be unaffected by the galaxy's magnetic field. Here we will focus on the photon spectrum of an EBH, which may be observed by a gamma ray observatory.

Primary photons are radiated directly from the EBH, according to eq. (2). The other particles which are radiated may produce secondary photons, as final state radiation or as the particles hadronise and decay. The secondary photon spectrum is given by

the sum of the primary spectra integrated against the secondary spectrum of a primary particle $i$ with energy $E_p$, $dN^{i\to\gamma}/dE$,

$$\frac{d^2N_s^\gamma}{dtdE} = \sum_{i\neq\gamma}\int_0^\infty \frac{d^2N_p^i}{dtdE_p}(M,E_p)\frac{dN^{i\to\gamma}}{dE}(E_p,E)dE_p\,.$$
(5)

Computation of the secondary photon spectra is relatively complex, particularly in the case of coloured particles which hadronise. To calculate the secondary spectra we use the public code `Pythia 8.3` [78]. The secondary photon spectra for several BH masses are shown in fig. 1.

Once produced, these photons then travel to the earth where they may be detected. The number of photons reaching the earth per $m^2$ will be reduced by the geometric factor $1/4\pi r^2$, where $r$ is the distance to the EBH. Although an EBH is yet to be observed, we investigate what information could be obtained if one were to be seen in a ground-based gamma ray observatory. As an illustrative example we take HAWC, the High Altitude Water Cherenkov Experiment located in Mexico at an altitude of 4100 meters, which started running in 2015. For gamma rays above $10^7$ GeV, HAWC has an effective area of $\sim 10^5\,m^2$, but this falls off sharply at lower energies; at $100$ GeV it is just $\sim 50\,m^2$. The parameterisation of the effective area can be found in ref. [79]. Although we expect very few photons to be observed above $10^5$ GeV, we extrapolate the effective area from $10^5$ GeV to $10^7$ GeV, with a constant effective area.

In addition to any photons from an EBH, HAWC will observe photons from the astrophysical gamma ray background. The Fermi-LAT collaboration has measured the isotropic diffuse gamma ray background, and we use their model A parameterisation [80] to account for this. Due to HAWC's good angular resolution (better than 2 degrees at all energies), we expect less than one background event in $10^6$ s of observation.

**Probing the Dark Sector** – To illustrate the sensitivity of an observation to BSM physics, we take the example of a dark sector (DS). As it is not known whether the DS communicates with the SM via interactions beyond the gravitational interaction, it is very difficult to conclusively probe these models in conventional dark matter experiments. However, since Hawking radiation is independent of these couplings, EBHs are uniquely placed to shine a light on the DS.

The DS could be simply a single dark matter particle, DS($\chi$) where we take $\chi$ to be a Dirac fermion,

or could contain many more degrees of freedom, see e.g., refs. [81–88]. For illustrative purposes we consider models motivated by the Mirror Dark Matter [83] scenario, where the DS contains an exact copy of the SM degrees of freedom which communicate with the SM only via small portal couplings. Generalising [83], we will assume $N$ copies of the SM and take all particles in the dark sector to have a common mass, $\Lambda_{DS}$. We will denote these models DS($N,\Lambda_{DS}$). The function $\alpha$ for two benchmark models are shown in fig. 2. The increase in $\alpha$ at black hole masses $\sim 10^{10}\,(10^{12})$ g leads to an accelerated evaporation rate in the final $\sim 10^3\,(10^9)$ s of the BHs life. Since the DS particles will produce no (or very few) secondary photons, this acceleration will indicate the existence of the DS.

To distinguish SM evolution from BSM evolution at the HAWC observatory, we integrate the total photon spectra against the HAWC effective area over all energies and over intervals in the remaining lifetime of the EBH, $\tau$,

$$N_j = \frac{1}{4\pi r^2}\int_0^\infty dE \int_{\tau_j}^{\tau_{j+1}} d\tau \frac{d^2N_{p+s}^\gamma}{d\tau dE}A(E,\theta,\tau)\,,\ (6)$$

where $A(E,\theta,\tau)$ is the effective area at zenith angle $\theta$ and time $\tau$, and $\tau_j \in \{10^{-4},10^{-2},10^0,10^2,10^4,10^6\}$.[1] While this approach does not make use of the photon energy spectrum, we note that HAWC's energy resolution is relatively poor ($\sim 50\%$ for photons above $10^4$ GeV). It does however make good use of the timing information, where HAWC has excellent resolution (order $100$ ps). To approximate the motion of the EBH through the sky, we assume that the HAWC detector lies on the equator of the earth (it in fact lies at $19°$ N) and that the EBH occurs on the celestial equator. We also assume that the EBH spends its final $\sim 3$ hours in the primary zenith angle band ($-26°$ to $26°$). We model the prior passage through the zenith bands given in [79] as the earth rotates.

The integrated photon counts for the SM and two benchmark DS models are shown in fig. 3, for an EBH seen at a distance of $0.015$ pc. We see that more degrees of freedom lead to a lower photon count, due to the accelerated evaporation rate. We also see that

--------

[1] We do not use times shorter than $10^{-4}$ s as below this time the BHs may produce primary particles with $E > 10^7$ GeV, and `Pythia 8.3` only produces reliable results below $E \sim 10^7$ GeV. A further bin up to $10^8$ s would contain significant astrophysical background.

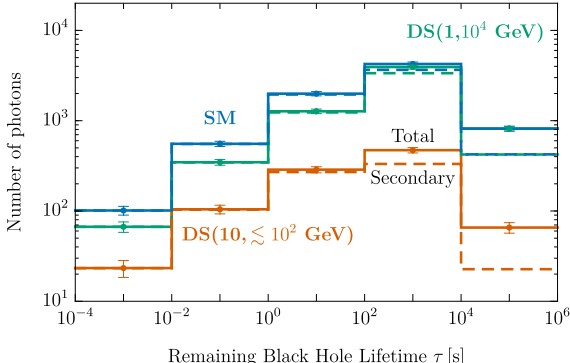

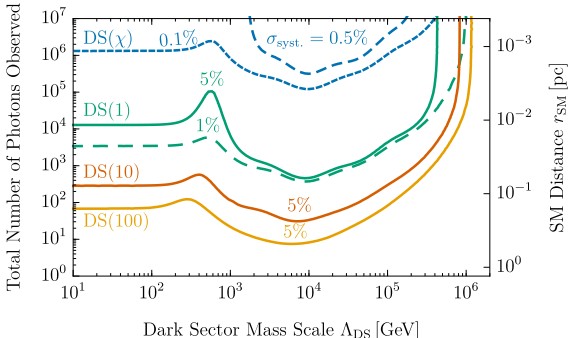

FIG. 3. The total (solid) number of photons observed in each time window for an EBH observed at 0.015 pc for the SM and two dark sector models. The secondary photon contribution is shown with a dashed line. The error bars include statistical and 5% systematic errors.

FIG. 4. Projected $2\sigma$ exclusion limits for a range of dark sector models, for different systematic errors. The search assumes that a given total number of photons is observed between $10^{-4}$ and $10^6$ s, with a SM-like spectrum. The distance to the EBH, assuming only the SM, is given by the right axis.

a relatively light DS ($\Lambda_{\rm DS} \lesssim 10^2$ GeV) leads to a reduction in the spectrum at all times, while a heavier DS ($\Lambda_{\rm DS} \sim 10^4$ GeV) only alters the spectrum below $\tau \sim 10^2$ s. This is because the EBH is only hot enough to emit such heavy particles in its last 100 s. Primary photons are seen to make up a significant proportion of the observed photons at $\tau \gtrsim 10^4$ s, but are a negligible contribution for $\tau \lesssim 10^2$ s.

When an EBH is observed, however, its distance from earth will be unknown. If the SM is assumed, the total photon count can be used to determine the distance. Since here we are constraining BSM models, we cannot make this assumption. Instead, we characterise the event by the number of photons observed between $10^{-4}$ and $10^6$ s. We normalise the SM and BSM spectra, such as those presented in fig. 3, to yield this total photon count. We then perform a chi-squared test between the expected observed spectrum (given by the SM) and the BSM spectra. We add the statistical and systematic errors in each bin in quadrature.

Figure 4 shows the expected $2\sigma$ limits that could be placed on various DS models for different DS mass scales and different systematic errors. The left axis gives the total number of photons observed between $10^{-4}$ and $10^6$ s, while the right axis gives the inferred distance to the EBH assuming only the SM. If the local EBH density is near the current upper limit [89], the probability of HAWC observing at least one event in the next five years at a distance less than 0.05 (0.01) pc is $\sim 83\%$ (1.4%).

We see that when there are more degrees of freedom in the DS, fewer photons are required to exclude the model. DS(100) can be essentially excluded up to $10^5$ GeV with just 100 photons, while DS(1) requires

$\sim 10^4$ photons to exclude mass scales $\lesssim 3 \times 10^5$ GeV.

For a dark sector mass scale $\lesssim 100$ GeV, the new radiation processes have fully opened by $\tau \sim 10^6$ s. Since this is the total length of assumed observation time, the search becomes independent of the mass scale below $\sim 100$ GeV. However, at mass scales $\gtrsim 10^6$ GeV, the search loses sensitivity since the EBH only emits such high mass particles at $\tau \lesssim 10^{-4}$ s.

In the top half of the plot, so many photons are received that the systematic error has a significant impact on the limit. We see that $\sigma_{\rm syst.} \lesssim 0.5\%$ is required to place limits on the DS($\chi$) model. In the lower half of the plot, the exclusion limit is dominated by statistical errors and the limit does not significantly change for $\sigma_{\rm syst.} \lesssim 5\%$.

**Conclusions** − The observation of an EBH can place significant constraints on the number of elementary degrees of freedom present in nature. We have exemplified this with a variety of dark sector models, and found that the number of new degrees of freedom below $\sim 3 \times 10^5$ GeV could conceivably be limited to less than one copy of the SM degrees of freedom in the near future.

The approach outlined here could readily be extended to further BSM models, in particular those with large numbers of new degrees of freedom. Given that such an observation can probe mass scales up to $\sim 10^6$ GeV,[2] models which address the hierarchy problem, such as SUSY, composite Higgs mod-

---

[2] While we have demonstrated sensitivity to models with mass scales below $\sim 10^6$ GeV, this limit is somewhat artificial and stems from our inability to accurately model the secondary spectra above $10^7$ GeV. The experimental timing resolution allows for measurement down to $\sim 100$ ps, in

els and NNaturalness [90], would be of particular interest. Other interesting scenarios would be light new physics sectors, where the non-gravitational interaction strengths are typically very weak, or further models with large numbers of new particles such as extra dimensional models with towers of KK resonances or string theory (which often leads to an abundance of light scalar particles). In contrast to the dark sector models considered here, some of these new particles will produce additional secondary photons, which may improve the sensitivity of both the initial EBH search and the information that can be extracted from the signal.

As a note of caution, the expected chance of observing an EBH in the near future remains uncertain. While the probabilities given above assume the upper limit of the local BH burst rate, limits from galactic and extragalactic physics are significantly stronger [44, 91, 92]. The situation remains unclear as these limits depend sensitively on various assumptions such as the degree of local clumping, the production and propagation of anti-protons, and the validity of the Standard Model.

However, as we have demonstrated, the information obtained from an observation would be unique and of fundamental importance. While we have considered five years of observation by the HAWC observatory, improved experiments such as LHAASO [93], CTA [94] and SGSO[95] are already running or are in development. The larger effective area of these experiments significantly increases the potential observation rate, and improved energy resolution could help determine the distance to an EBH even in BSM scenarios. Furthermore, multiple experiments could potentially observe the same event (at similar or lower photon energies), and multi-messenger approaches could possibly see the event in other particles, such as neutrinos.

**Acknowledgements** – The authors would like to thank Peter Skands for advice on using Pythia 8.3, and would like to acknowledge support from the Australian Government through the Australian Research Council.

---

* michael.baker@unimelb.edu.au

† andrea.thamm@unimelb.edu.au

---

principle probing masses up to $\sim 10^8$ GeV (although the photon count will continue to reduce in shorter time windows).

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

*Dark Matter: Recent Developments*, *Ann. Rev. Nucl. Part. Sci.* **70** (2020) 355 [`2006.02838`].

*JCAP* **04** (2018) 009 [1712.07664].

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

[74] *Microscopic Black Holes*, *Phys. Rev. D* **73** (2006) 124024 [astro-ph/0604439].

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
