# Peer review of "Probing the Particle Spectrum of Nature with Evaporating Black Holes"

_SciPost Physics_

## Round 1 · Referee Report · Anonymous (Referee 1) · 2021-11-24

Strengths

1) The presented analysis appears to be generally valid from a technical standpoint, follows known methodology.

2) The authors extend previous results on observations of evaporating PBHs.

Weaknesses

1) There appear to be more general conceptual issues about feasibility of such analysis, in particular the probability and rate estimates of local PBH encounters are lacking. The authors dismiss existing strong constraints if such PBHs contribute to the dark matter.

2) The study is quite limited in the scope of the PBH emission signatures the authors consider, focusing on HAWC and a quite specific set of assumptions.

Report

Primordial black holes from the early Universe offer unique opportunities to explore new physics and this is a topic of active research. Emission spectrum from evaporating PBHs is known to be sensitive to the details of considered underlying particle model. The authors study detection prospects of emission from evaporation of a would-be nearby PBH as a method for probing dark sectors.

Considerations that particle models and dark sectors can impact PBH evaporation spectrum and hence observations have been previously discussed in variety of contexts, just one example being Masina Ref. [61] the authors themselves mention (albeit in case of early evaporation). So I am not sure it is fair to say that impact of BSM models in this context is "almost completely unexplored", as the authors claim. To analyze emission from a local nearby PBH, the authors follow standard methodology.

There are substantial issues with this work that I think need to be addressed. Most importantly, the whole study relies on probable detection of abundance of local PBHs. The authors do not seem to provide supporting estimates for this. This can be accomplished if PBHs constitute a fraction of dark matter. However, such PBH DM is already strongly constrained and the authors explicitly seem to say they completely ignore such limits.

The authors also seem to only focus on detection prospects for HAWC, and for a quite particular set of assumptions. Fermi Collaboration for example also did similar analyses (arXiv:1802.00100), which do not seem to be mentioned. I also wasn't able to find estimates about e.g. neutrino detection.

Requested changes

1) The authors should provide estimates and arguments for detection probability and event rate of such local PBHs in the setup they consider.

2) The authors mention BlackHawk code, but do not seem to use it. How do their results compare if they are to use BlackHawk emission output, which as I understand includes e.g. primary and secondary contributions and hadronization?

3) What are detection prospects for Fermi, and also for neutrinos?

  • validity: good
  • significance: ok
  • originality: low
  • clarity: good
  • formatting: excellent
  • grammar: excellent

Author:  Michael J. Baker  on 2022-02-15  [id 2203]

(in reply to Report 1 on 2021-11-24)

We thank the referee for their useful comments and questions, to which we reply as follows:

Requested changes:

  1. We have provided a more detailed account of the observational constraints in our new 'Discussion’ section. While there are a range of possible constraints, we believe that none of them are currently secure enough to conclude that an EBH will not be observed in the near future. We would also like to point the referee to our estimates in our 'Probing the Dark Sector’ section: “If the local EBH density is near the current upper limit ($3400 \,\text{pc}^{-3}\,\text{yr}^{-1}$), the probability of HAWC observing at least one event in the next five years at a distance less than $0.05$ $(0.01)\,\text{pc}$ is $\sim 83\%$ $(1.4\%)$, which corresponds to observation of around 10 (200) photons”

  2. We used the tabulated values of the greybody factors included with the BlackHawk code, and validated our primary and secondary spectra against the BlackHawk output. We didn’t simply use the BlackHawk output as we identified several minor bugs in their implementation (and notified the authors). We have added clarifying comments following eqs. (2) and (5).

  3. We have introduced a more detailed comparison of a variety of instruments in the new `Discussion’ section. While Fermi-LAT has performed similar searches to HAWC and obtained similar limits, its small size of 1 m$^2$ means that it is not a good probe of the final burst of an evaporating black hole. Neutrino experiments could be interesting, especially when viewed as part of a multi-messenger approach. While this is beyond the scope of the current work, we think it would be very interesting to investigate in the future.

Other changes:

We have rephrased our statement in the introduction, to clarify the relationship between our work and related literature.

---

## Round 1 · Referee Report · Anonymous (Referee 2) · 2021-11-28

Strengths

  1. The work considers an interesting question
  2. The calculations presented appear sound
  3. For the most part, the manuscript is well written

Weaknesses

  1. The proposed search requires a black hole near the end of its lifetime very close to the Earth, and the requirements to achieve this are not particularly clearly discussed
  2. The results seem to be technically limited by an inability to compute spectra above 10 PeV, but a solution to this problem exists in the literature
  3. The description of several aspects of the background and experimental assumptions were unclear

Report

In the submitted manuscript, the authors consider what could be learnt from gamma-ray observations of an evaporating black hole near the location of the Earth. In particular, the work demonstrates how we could infer the presence of states beyond the Standard Model (SM) directly from these observations, which is particularly interesting as these states may only couple to the SM gravitationally, and thereby be extremely hard to detect otherwise.

Fundamentally, to perform such searches a nearby black hole that is near the end of its lifetime is required. The likelihood of such an event is briefly discussed at the end of the paragraph beginning “Figure 4 shows the expected…”, but I would agree with the other referee that a much more detailed description of this fundamental point is required. My impression is that under generic assumptions, such an event is very unlikely, although I think the study would be much clearer if this question was answered (and some sense for what usual assumptions must be relaxed and how that could be achieved is provided).

In order to determine the full spectrum of photons produced, the authors also include the secondary production of photons that result from showering and hadronization. This is achieved by using Pythia 8.3. However, as the authors note in footnote 1, this then leads to a technical limitation in their results as they cannot simulate energies higher than 10 PeV. Given the limitation, I am unsure why they chose to use Pythia for this work. As they are only making use of the photon energy spectrum, and not the full angular information of the event, there are other results they could rely on. For instance, I am aware of [Bauer+ 2007.15001] where fragmentation functions well above the electroweak scale were computed that I assume the authors could use. While I suspect those results can’t be used to arbitrarily high masses, as there will be large uncertainties for low energy fractions, these should at least allow the authors to move to the 100 ps limit they mention in footnote 2.

Finally, I found several aspects of the analysis procedure and experimental configuration hard to follow and would have appreciated a clearer description. While I do not believe an enormous amount of detail is required here given the optimistic nature of the signal, I still think several clarifications would be worthwhile and I note these in the requested changes below.

In summary, while a large degree of luck may be required to see the black hole signal the authors address in this work, the potential for discovery if we did would be enormous. As such, I think this is a very interesting question that the authors have considered, and if my concerns can be addressed I would recommend the work for publication.

Requested changes

Below I list my main comments, followed by several minor suggestions I leave to the discretion of the authors.

  1. A clearer and more detailed description of the likelihood for a black hole in the required mass range sufficiently close the Earth, along with a discussion of what usual assumptions may need to be relaxed (and how this could be achieved) in order to do so.
  2. As mentioned in the report, I would have thought the results of [Bauer+ 2007.15001] would be more useful for the calculations than Pythia, especially given the high energy limitation. If there is a technical reason this isn’t the case, this should be explained. In principle that work also considers electroweak effects missing in Pythia that could become relevant at the energies considered in this work, but my impression is those effects would only give a small correction (as most of the secondary photons arise from QCD effects which Pythia describes accurately).
  3. I found several aspects related to the analysis confusing, and would appreciate clarification. In particular:
    • The estimate of less than one background photon for the million second observation could be clarified. The authors use a Fermi model of the IGRB to infer the background for HAWC, but Fermi only sees photons up to 1 TeV, so clearly some extrapolation is required, and should at least be clearly stated. Further, this won’t be the only source of background in general. Unless the black hole is at high galactic latitudes, there can be galactic emission also, and further HAWC has a background from rejecting charged cosmic rays. Potentially the authors estimated both of these and found them negligible, but if so a statement as such would be useful.
    • More generally, I would have found a discussion as to why HAWC is the relevant instrument for such searches useful. Given the black holes will appear as point sources, I would have naively imagined an instrument like HESS (or in the future CTA) which has improved angular resolution to be helpful. Possibly the point is that the backgrounds are so low that the additional background rejection one gets from the improved localization is irrelevant? These instruments also have better energy resolution, but for the analysis envisioned by the authors this may not be relevant. Further, especially if the analysis is extended to higher energies I wonder if neutrino instruments could weigh in, although again the photon channels may simply have higher sensitivity.
    • Given that LHAASO is already operating a subset of its array, how much it is expected to improve such an analysis would have been interesting to at least sketch.

Minor comments: 1. In Fig. 1, I would have appreciated the temperature corresponding to at least one of these masses to be provided (the other temperatures can then be inferred from Eq. (1)). 2. I would have found a brief discussion of what impact the greybody factors have on their results useful. 3. The effect will be small, but technically in (5) the sum should also run over photons, as the photon itself has a non-trivial fragmentation function. 4. At the energies considered in this work, photons develop an optical depth due to conversion to e+ e- pairs off various radiation fields (including the CMB). For the distances considered, I believe the optical depth is negligible, but the authors may wish to mention this (see, for example, Fig 1 or Fig. 3 of [Esmaili+ 1505.06486]). 5. I assume the authors use the full HAWC energy range, but explaining what that range is (at least where the instrument has appreciable effective area) would have been helpful. 6. For the estimated distance of the black hole, the approach of the authors seems to work in the simple model considered, where all particles have a single mass scale, but I worry in a more general model this would become more challenging. While it comes with its own assumptions (in particular the absence of additional light degrees of freedom beyond the standard model), I wonder if calibrating the distance off the spectrum for temperatures below the electroweak scale (and for lifetimes beyond a million seconds) is plausible.

  • validity: good
  • significance: ok
  • originality: good
  • clarity: good
  • formatting: perfect
  • grammar: perfect

Author:  Michael J. Baker  on 2022-02-15  [id 2202]

(in reply to Report 2 on 2021-11-28)
Category:
answer to question

We thank the referee for their useful comments and questions, to which we reply as follows:

Main comments

  1. We have provided a more detailed account of the observational constraints in our new `Discussion’ section. While there are a range of possible constraints, we believe that none of them are currently secure enough to conclude that an EBH will not be observed in the near future.

  2. We have rephrased footnote 1 and introduced a paragraph to the new `Discussion’ section, clarifying that Pythia is sufficient for our analysis but that it may be better for future analyses to use software designed to work at high scales, such as [Bauer+ 2007.15001].

  3. We thank the referee for asking us to look again at the backgrounds. Misidentified cosmic rays are in fact the dominant background. We have altered our analysis to take account of this, introducing an energy cut and reducing the observation time to retain a background of less than one event. While the reduced observation time reduces the sensitivity of our search, the energy cut improves it, so the final results are not greatly altered.

  4. We have added several sentences to the end of the `Formalism’ section, describing these backgrounds and our extrapolation procedure.

  5. We have introduced a more detailed comparison of a variety of instruments in the new `Discussion’ section. While HESS and CTA are better than HAWC in some regards, they have a much smaller field of view. As such, they are much less likely to observe an evaporating black hole by chance. It is not clear how effective early warning systems would be for this signal, and this would be something interesting to investigate in the future. Neutrino experiments could also be interesting, especially when viewed as part of a multi-messenger approach. While this is beyond the scope of the current work, we think it would be very interesting to investigate in the future.

  6. While in many respects LHAASO and HAWC are comparable, LHAASO has a significantly lower cosmic ray background above 10 TeV. However, as we write in the new `Discussion’ section, it is not clear that this will lead to an improvement over HAWC.

Minor comments

  1. We have added the temperature corresponding to one mass, and point out that for photons the greybody factor significantly suppresses the emission around $E \sim T$ (as can be seen from the fact that the primary photon spectrum peaks at an energy almost an order of magnitude larger than the black hole temperature).

  2. We have added some clarifying sentences after eq. (2). We also checked numerically that this is what happens in our code when we omit the greybody factor.

  3. We model the evaporating black hole as only emitting on-shell photons, which do not decay. While we believe any off-shell photon emission will be significantly suppressed and can safely be neglected, we would welcome any further discussion on this point.

  4. This effect is indeed negligible for our study, but we have added a sentence to the `Formalism’ section.

  5. The full energy range is approximately 100 GeV to above $10^5$ GeV, but HAWC only has an effective area of $10^5$ m$^2$ above $10^4$ GeV (we have fixed a typo where we wrote $10^7$ GeV instead of $10^4$ GeV). We have added this information to the `Formalism’ section.

  6. We only estimate the distance of the black hole assuming the SM. If BSM particles are present, we can not estimate the distance and the distance scale in Figure 5 does not apply. We have introduced footnote 2 to clarify this point, and have added the referee’s suggestion of calibration using information from the first part of the signal.

---

## Editorial Decision

resubmitted